# *Trichoderma reesei* Contains a Biosynthetic Gene Cluster That Encodes the Antifungal Agent Ilicicolin H

**DOI:** 10.3390/jof7121034

**Published:** 2021-12-01

**Authors:** Mary L. Shenouda, Maria Ambilika, Russell J. Cox

**Affiliations:** 1Institute for Organic Chemistry and Biomolekulares Wirkstoffzentrum (BMWZ), Schneiderberg 38, 30167 Hannover, Germany; mary.shenouda@oci.uni-hannover.de (M.L.S.); m.ambilika@gmail.com (M.A.); 2Pharmacognosy Department, Faculty of Pharmacy, Alexandria University, Alexandria 21521, Egypt

**Keywords:** heterologous expression, PKS-NRPS, ilicicolin H, shunt metabolite, *Trichoderma reesei*

## Abstract

The *trili* biosynthetic gene cluster (BGC) from the well-studied organism *Trichoderma reesei* was studied by heterologous expression in the fungal host *Aspergillus oryzae*. Coexpression of *triliA* and *triliB* produces two new acyl tetramic acids. Addition of the ring-expanding cytochrome P450 encoded by *triliC* then yields a known pyridone intermediate to ilicicolin H and a new chain-truncated shunt metabolite. Finally, addition of the intramolecular Diels-Alderase encoded by *triliD* affords a mixture of 8-epi ilicicolin H and ilicicolin H itself, showing that the *T. reesei trili* BGC encodes biosynthesis of this potent antifungal agent. Unexpected *A. oryzae* shunt pathways are responsible for the production of the new compounds, emphasising the role of fungal hosts in catalysing diversification reactions.

## 1. Introduction

*Trichoderma reesei* (*Hypocrea jecorina*) is a filamentous fungus widely used for the industrial production of cellulase [1]. The organism can grow to high cell density and can grow robustly on a variety of low-quality substrates. It has been subject to a wide variety of biotechnological improvements since the discovery of the wild-type QM6A strain in the 1940s, and many molecular tools are available for its genetic manipulation. Many *Trichoderma* species have been used in the remediation of fungal diseases. For example, Foster and coworkers have shown that some *Trichoderma* spp. can effectively treat *Armillaria mellea* (honey fungus) that causes root rot in many tree species [2]. *T. reesei* is also well-known for the production of a number of secondary metabolites such as sorbicillinol **1** [3,4], and numerous multimeric derivatives such as bisvertinolone **2** [5].

Other members of the *Trichoderma* genus are well-known producers of many secondary metabolites [6]. In particular, compounds such as acyl tetramic acids (acyl 2-pyrollones) and acyl 2-pyridones, synthesised by hybrid polyketide synthase-nonribosomal peptide synthetases (PKS-NRPS) are commonly produced by various *Trichoderma* species. These include metabolites such as harzianic acid **3** (*T. harzianum*) [7,8], and deacetyl-18-deoxy cytochalasin H **4** [9] among the pyrollones; and trichodin A **5** [10] and harzianopyridone **6** [11,12] among the 2-pyridones (Figure 1). The biosynthesis of related pyrollones such as fusarin C **7** in *Fusarium* spp [13] and acurin A **8** in *Aspergillus aculeatus*, [14] and 2-pyridones such as tenellin **19** in *Beauveria bassiana* [15] and ilicicolin H **10** in *Talaromyces variabilis* and *Nectria* spp [16,17] are well-understood.

However, despite decades of study, *T. reesei* itself has not been reported to produce any PKS-NRPS compounds. Since we have a broad interest in fungal PKS-NRPS compounds and we recently made detailed studies on the biosynthesis of sorbicillinoids in *T. reesei* [3,4], we set out to determine if this organism can also produce PKS-NRPS compounds.

## 2. Materials and Methods

All chemicals and media ingredients used in this work were purchased from Duchefa Biochemie, Roth, VWR, Fisher scientific, Sigma Aldrich, abcr and Formedium. All media, buffers, solutions and antibiotics were prepared with double-distilled water (ddH_2_O), where deionized water was further purified by GenPure Pro UV/UF Milipore device from Thermo scientific. All growth media and sterile solutions were autoclaved at 121 °C for 15 min using Systec VX150 or a classic Prestige Medical 2100 autoclave. Antibiotic solutions were sterilized using a sterile syringe filter (0.45 µm pore size, Roth). See Appendix A for details of all growth media, buffers and solutions, enzymes and antibiotics.

### 2.1. Microbiological Methods

Bacterial and fungal strains and all microbiological methods used in this work are summarized in the Electronic Appendix A.

### 2.2. Construction of Heterologous Expression Vectors

The genomic DNA (gDNA) of *T. reesei* QM6a *∆tmus53 ∆pyr4* [18] was used as a template to amplify genes for yeast homologous recombination (without removal of introns). The vector pEYA was used to assemble the PKS-NRPS (*TRIREDRAFT_58285, triliA*) gene (~12 kb) in four different DNA fragments of around 3 kb each using yeast homologous recombination. This was followed by LR recombination between the entry plasmid pEYA·*triliA* (MSIII139) and the expression vector pTYGS·*arg* to yield vector pTYGS·*argB*·*triliA* (MSIII144). The enoyl reductase gene (*TRIREDRAFT_58289, triliB*) was also amplified from the gDNA of *T. reesei* using primers with overhang to *P_gpdA_* and *T_gpdA_*. Using yeast homologous recombination, the final vector pTYGS·*argB*·*triliA·triliB* (MSIII152) was constructed which contains the PKS-NRPS (*triliA*)- and ER (*triliB*)-encoding genes from the specified cluster. A vector for the expression of the first three genes of the cluster; PKS/NRPS (TRIREDRAFT_58285, *triliA*), ER (TRIREDRAFT_58289, *triliB*), P450_RE_ (*TRIREDRAFT_58953, triliC*) was constructed using yeast homologous recombination using gDNA of *T. reesei* QM6a·*∆tmus53*·*∆Pyr4* as a template for the amplification of the genes. A vector for the expression of the first four genes of the cluster; PKS/NRPS (TRIREDRAFT_58285, *triliA*), ER (TRIREDRAFT_58289, *triliB)*, P450_RE_ (*TRIREDRAFT_58953, triliC*) and *Diels-Alderase* (*triliD*) was constructed via yeast homologous recombination using gDNA of *T. reesei* QM6a·*∆tmus53*·*∆Pyr4* as template for the amplification of the genes.

### 2.3. Chemical Analysis

All the chemicals and materials were purchased from one of the following companies: Bio-Rad (München, Germany), New England Biolabs (Beverly, MA, USA), Roth (Karlsruhe, Germany), Sigma Aldrich (Steinheim, Germany), and Thermo Fisher Scientific (Waltham, MA, USA).

#### 2.3.1. Liquid Chromatography–Mass Spectrometry (LC–MS)

Analytical LC–MS data of the organic extracts were obtained by either a Waters mass-directed autopurification system (Waters, Wilmslow, UK) including a Waters 2767 autosampler, a Waters 2545 pump and Phenomenex Kinetex column (2.6 μm, C_18_, 100 Å, 4.6 mm × 100 mm; Phenomenex) equipped with a Phenomenex Security Guard precolumn (Luna, C_5_, 300 Å). Detection was performed by a diode array detector from 210–600 nm (DAD; Waters 2998 or Waters 996), an electron light-scattering detector (ELSD; Waters 2424) and an electrospray ionisation mass detector in the range of 100–1000 *m/z* (Waters SQD-2). Gradient was run over 15 min starting at 10% acetonitrile/ 90% HPLC grade water (+ 0.05% formic acid) and ramping to 90% acetonitrile/ 10% HPLC grade water (+ 0.05% formic acid). Flowrate was 1 mL/min and 20 μL of the sample was injected. Data were displayed using the software MassLynx. The LC–MS program is summarized in the Appendix A.

Single compounds were purified from the raw organic extracts by a Waters mass-directed autopurification system. It comprises a Waters 2767 autosampler, a Waters 2545 binary gradient module system and a Phenomenex Kinetex Axia column (5 μm, C_18_, 100 Å, 21.2 mm × 250 mm) equipped with a Phenomenex Security Guard precolumn (Luna, C_5_, 300 Å). A water/acetonitrile gradient was run over 15 min with a flowrate of 20 mL/min and a post-column flow split of 100:1. The minority flow was applied for simultaneous analysis by a diode array detector (Waters 2998) in the range 210 to 1600 nm, an evaporative light-scattering detector (ELSD; Waters 2424) and a Waters SQD-2 mass detector, operating in ES+ and ES− modes between 100 and 1000 *m/z*. Selected peaks were collected into test tubes and solvent was evaporated under reduced pressure. Compounds were dissolved in methanol (1 mg/mL). High-Resolution Mass Spectrometry was performed on a Q-Tof Premier mass spectrometer (Waters) coupled to an Acquity UPLC–domain (Waters). Electron spray Ionisation (EsI) mass spectroscopy was measured in positive or negative mode depending on the compound.

#### 2.3.2. Nuclear Magnetic Resonance (NMR) Analysis

Bruker Ascend 400 MHz, Bruker DRX 500 MHz or a Bruker Ascend 600 MHz Spectrometer (Bruker) were used for NMR measurements of the samples. Raw data were then analyzed using the software Bruker TopSpin 3.5. Chemical shifts are expressed in parts per million (ppm) in comparison to the Tetramethylsilane (TMS) standard and are referenced to the deuterated solvent.

#### 2.3.3. Extraction of Fungal Cultures

For *A. oryzae* transformants, a small cell sample was dried by Büchner filtration or gravity filtration and used for gDNA analysis. Mycelia were ground using a hand blender and removed from the culture supernatant by Büchner filtration. Supernatant was acidified with 2 M HCL to pH 3–4 and extracted twice with ethyl acetate. Combined organic layers were dried over MgSO_4_ and solvents were removed under vacuum. Organic residue was dissolved in methanol or acetonitrile to a concentration of 5–10 mg/mL (analytical) or 50 mg/mL (preparative) and filtered over glass wool.

## 3. Results

The genome of *T. reesei* QM6A, the wild-type isolate, was sequenced in 2008 [19]. We examined this genome sequence using a combination of automated AntiSMASH [20] and manual screening, with a particular focus on PKS and PKS-NRPS encoding genes. Bioinformatic analysis of all PKS and PKS-NRPS genes in *T. reesei* QM6a revealed the presence of 11 PKS and two hybrid PKS-NRPS encoding genes (see Appendix A for details). Of these, only the two PKS genes responsible for the sorbicillins and related compounds have been identified and confirmed (TRIREDRAFT_73618 and 73621) [4,21,22]. The HR-PKS TRIREDRAFT_65116 shows a very high similarity to the recently investigated PKS from *Trichoderma virens,* which is encoded in the biosynthetic gene cluster (BGC) that directs the production of virensols and trichoxide [23]. However, the other PKS and PKS-NRPS BGCs could not be linked to the production of any specific compound.

Although the *T. reesei* genome contains two hybrid PKS-NRPS encoding genes, no PKS-NRPS secondary metabolites have been reported from this fungus and these BGCs are likely to be silent under laboratory conditions. The two PKS-NRPS themselves show similarity (>60%) to proteins involved in fusarin C **8** biosynthesis in *Fusarium moniliforme* [13] and illicicolin H **10** biosynthesis in *Penicillium variabile* (= *Talaromyces variabilis*) [16]. Ilicicolin H **10** is an interesting antifungal agent that targets the cytochrome bc1 respiration complex in the 3–5 nM range [24], and it is conceivable that ilicicolin H **10** production may be responsible for the observed antifungal effects of *Trichoderma* species in general [2] and *T. reesei* in particular [25]. We therefore selected the putative *T. reesei* ilicicolin H BGC (*trili)* for further investigation.

Based on the NCBI prediction, the *trili* BGC (Figure 2) consists of a hybrid PKS/NRPS (*triliA*), an ER (*triliB*), and a putative cytochrome P450 (*triliC*). Reannotation of the cluster by FungiSMASH and Clinker [26] and Artemis comparison to a closely related BGC in *T. citrinoviride* resulted in the identification of a putative *S*-adenosylmethionine (SAM)-dependent Diels-Alderase in the cluster (*triliD*), that shows high similarity to *iccD* from the illicicolin-H BGC in *Talaromyces variabilis* [16]. Finally, *triliE* encodes a putative TIM-like epimerase homologous to IccE, known to epimerise epi-ilicicolin H in *T. variabilis*. Multigene analysis using cblaster [27] identified highly similar *ili* BGCs in other strains of *T. reesei* (RUT C-30) and other *Trichoderma* species, including *T. parareesei*, *T. virens* Gv29-8, *T. citrinoviride* TUCIM6016 and *T. longibrachiatum* ATCC 1864*8* (see Appendix A).

In an initial heterologous expression experiment, *triliA* and *triliB* were cloned into the vector pTYGS·*argB* [28] using rapid recombination of genome-derived PCR fragments in yeast and *in vitro* (Gateway) recombination. The PKS-NRPS-encoding gene *triliA* was placed downstream of the inducible *A. oryzae amyB* promoter (*P_amyB_*), while *triliB* was placed under the control of the *Aspergillus gpdA* promoter (*P_gpdA_*). The resulting vector pTYGS·*argB*·*triliA*·*triliB* was transformed into the fungal host *Aspergillus oryzae* NSAR1 [29]. Comparison of the organic extracts of transformants with those of untransformed *A. oryzae* NSAR1 revealed the production of new compounds by LC–MS analysis eluting at 6.7 min, 8.0 min and 11.0 min (Figure 3).

Compound **11** (*RT* 6.7 min, 30 mg) was isolated by preparative LC–MS, and the structure was elucidated using 1D and 2D NMR. HRMS analysis of **11** confirmed a molecular formula of C_23_H_28_NO_6_ ([M+H]+ calculated 414.1917, found 414.1917). Compound **12** (*RT* 8.0 min, 4 mg) was isolated by preparative LC–MS and the structure was elucidated using NMR. HRMS analysis of **12** confirmed a molecular formula of C_27_H_32_NO_6_ ([M+H]+ calculated 466.2230, found 466.2237).

NMR analysis of these compounds (see Appendix A for full details) showed that **11** is a linear acyl tetramic acid with a C_12_ sidechain terminating in an unusual (at this position) carboxylic acid. Minor compound **12** was shown to be a related acyl tetramic acid featuring a decalin, with an unusual 3-propenoic acid substituent. Although the minor compound **13** at 11.0 min could not be fully identified by NMR, mass spectroscopic and UV analysis suggested that it is likely to be a third related tetramic acid with a C_16_ sidechain, first reported by Tang and coworkers during biosynthetic studies of ilicicolin H (Figure 1) [16].

In the next round of experiments, *triliC*, encoding a cytochrome P450 oxidase, was added to the *A. oryzae* strains, driven by *P_adh_*. TriliC is related to ring-expanding cytochrome P450 oxidases first characterised from the tenellin pathway [15]. Coexpression of *triliA*, *triliB* and *triliC* resulted in production of two new compounds in comparison to the previous transformants and the untransformed *A. oryzae* (Figure 4), eluting at 6.7 (**14**), and 10.8 (**15**) min. Full NMR analysis of the *RT* 6.7 min compound showed it to be the pyridone homolog of **11**. Similarly, the 10.8 min compound was shown to be the pyridone homolog of **13**.

Finally, a gene homologous to Diels-Alderase *iccD* (*triliD*) was added to the expression system downstream of *P_eno_* and transformed into *A. oryzae*. This experiment again produced **11** (*RT* 6.7 min). Additionally, two new peaks in comparison to the previous transformants and the untransformed *A. oryzae* were also observed by LC–MS (Figure 5) at *RT* 10.76 and 11.0 min. The minor compound at 11.0 min was again assigned as **13** after analysis of partial data. NMR and HRMS analysis (see Appendix A for full details) showed the peak at 10.76 min consists of an almost coeleuting mixture of the known 8-epi ilicicolin H **16** and ilicicolin H **10** itself.

## 4. Discussion

PKS-NRPS compounds have been reported before from the genus *Trichoderma*, but *T. reesei* QM6A itself has not been reported to produce these compounds. Genomic analysis, however, reveals two typical fungal PKS-NRPS BGCs. Heterologous expression experiments reveal that the *trili* BGC is fully functional, but apparently silent in *T. reesei* QM6A under laboratory conditions. Initial expression of the PKS-NRPS (*triliA*) and *trans*-ER (*triliB*) genes in *A. oryzae* led to production of two new acyl tetramic acids (**11** and **12**) and a third compound **13** that is likely to be a precursor of illicicolin H, recently described by Tang and coworkers [16] from *Nectria sp. B13* and *T. variabilis* and Gao and coworkers [17] in *Neonectria sp. DH2*. Decalin **12** appears to have been oxidised to a carboxylic acid at its terminal polyketide methyl group. Addition of the ring-expanding P450 oxidase then produces the corresponding 2-pyridones **14** and **15**. Compound **15** is identical to an ilicicolin H precursor previously identified by Tang and coworkers [16], but **14** is new. Finally, addition of the Diels-Alderase encoded by *triliD* produces a mixture of 8-epi ilicicolin H **16** and ilicicolin H **10** itself.

The *T. variabilis icc* [16] and *Neonectra DH2 ili* [17] BGCs were previously investigated by expression in *A. nidulans*. Our results are highly similar to those obtained from these studies, except that *A. oryzae* as a host appears to catalyse some unexpected steps. For example, in *A. oryzae* the initial tetramic acid precursor **13** appears to be oxidatively degraded to remove the terminal four polyketide carbons and leaves a carboxylic acid in the structure of **11**. Previous heterologous expression studies on *A. oryzae* also indicate that this host has a propensity for the oxygenation of polyenes [30,31] and oxidative cleavage of olefins [32]. Likewise, oxygenation of the terminal methyl of **12** to a carboxylic acid is newly observed in *A. oryzae*. This is similar to the BueE-catalysed oxidation recently observed by Piggott and coworkers in the case of the fungal polyketide decalin burnettiene A in *Aspergillus burnettii* [33]. It also appears that a spontaneous (or *A. oryzae*-catalysed) Diels–Alder reaction can occur to produce a decalin that is then oxidised to a carboxylic acid at the terminal methyl to generate new compound **12**. In previous studies on ilicicolin H biosynthesis, using *A. nidulans* as the host, a specific Diels–Alder catalyst was required for this step. We also observed the chain-truncated pyridone **14**—again a new natural product. In *A. nidulans* it appears that 8-epi ilicicolin H **16** is a stable compound, requiring expression of *iccE* for the conversion to ilicicolin H. However, in *A. oryzae,* observable amounts of ilicicolin H **10** are produced in the absence of this epimerase catalyst.

Our results therefore illustrate that *T. reesei* QM6A has a fully functional PKS-NRPS BGC that encodes the biosynthesis of ilicicolin H, a potent antifungal agent. The ability of *T. reesei* QM6A to make this compound has never before been observed [34], despite nearly 70 years of laboratory-based investigations. It is likely that this BGC can be activated under unknown environmental conditions in its native habitat and may contribute to its utility as an agricultural antifungal agent, but it appears to be silent under laboratory conditions. Finally, our results also show that different fungal heterologous hosts can introduce unexpected shunt pathways which can diversify known secondary metabolites and produce new-to-nature compounds.

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
