# Peer review of "Trichoderma reesei Contains a Biosynthetic Gene Cluster That Encodes the Antifungal Agent Ilicicolin H"

_jof, 2021, doi:10.3390/jof7121034_

Round 1

Reviewer 1 Report

This paper started with investigating the T. reesei Genome, and focused on hybrid PKS-NRPS in T. reesei. Heterologous expression in A. oryzae was used to activate the silent genes cluster. Finally, authors predicted the biosynthesis pathway of the silent PKS-NRPS. This work is significative in confirming the specific compound, linked with predicted genes cluster in Trichoderma. I have a few minor suggestions for the authors to consider. Line 56. What is ESI? Line47-127 detailed methods in transformation in E. coli ,S. cerevisiae and A. aryzae is not required. Please derectly cite related literatures. Line 128- 172 please also simplify the plasmid extraction, PCR, and so on. This is not specific or novelty methods, what authors used. Line 265, Please unify the word font in Figure2. Combing figure2 and a table is muddled. Line 294, need to detail the mutants and compounds in figure3. It is difficult to understand the pictures information of LC-MS. Line 306 and 326. The same as above. Line 363 Authors only predict the pathway, please rescript Figure legends. The word in figure is small.

Reviewer 2 Report

This is a very nice concise study that used heterologous expression in Aspergillus to determine the functionality of Trichoderma reesei PKS-NRPS genes and the potential of this species to produce antifungal compounds.  It was clearly presented.  

a couple of typos:  

line 214 comprises

lines 222-223 compounds were dissolved

Reviewer 3 Report

The manuscript reports an interesting discovery of the trili biosynthetic gene cluster (BGC) encoding the antifungal agent ilicicolin H in the well-known filamentous fungus Trichoderma reesei. It appears that this BGC is silent and therefore its expression for a potential respective metabolite has never been detected. By the heterologous expression of this BGC in Aspergillus oryzae, the authors could detect the presence of the metabolite ilicicolin H. However, the manuscript should be improved for the presentation and by language proofreading. There are still numerous mistakes and unnecessary sentences in the submitted manuscript:

Line 26: spp. should be regular (not in italic form). 

Line 57: you can not write the title like this "Escherichia coli". It should be a protocol for E. coli transformation.

Line 65: mistake for "supernatant"

Line 68: similar to Line 57

Line 69: stain -> should be "strain"

Line 79: dd.H2O (deionized distilled water?)

Line 97: similar to Line 57, 68 (please also cite the reference for this strain)

Line 116-118: This information seems to be not convincing. You use the arginine auxotrophic strain (the gene for arginine is deleted), how the dead cell material from this strain maintains arginine to cause false-positive transformants.

Line 124: was -> should be "were"

Line 143: Sigma-Aldrich, where? USA, Germany, ...

Line 154: Provide the origin of the plasmids (citation?)

Line 157: Ebersberg, Germany ?

Line 159: this section (2.2.7.) should be removed. It is too basic.

Line 172: similar to Line 159

Line 179: gDNA --> should be "genomic DNA (gDNA)"

Line 180: (without removal of introns), this is redundant, because fungal gDNA usually contains introns. 

Lines 191-192: "using gDNA... of the genes" is redundant.

Lines 195-196: "using gDNA... of the genes" is redundant.

Line 203: similar to Line 143, 157 (where is the location of the company?)

Line 258: "illicicolin H 10 biosynthesis" is very confusing, you mean: "10" is compound 10. If it is like this, it should be "illicicolin H (compound 10) biosynthesis".

Lines 259-260: similar to Line 258, should consider defining compound 10 as C10.

Line 266: Figure 2 has a very confusing table. Please check carefully the accession numbers, "XP_" should be embedded in its codes, iliA = iccA?, iliC = iccC?

Lines 276-277: the names (RUT-C30, Gv-298, ...) of fungal strains should be regular (not italicized).

Line 280: remove this citation (just mention in the method section)

Line 284: remove this citation (just mention in the method section)

What about the other compounds (1-9)? Are they originated from A. oryzae?
